# Education for Sustainability in Practice: A Review of Current Strategies within Italian Universities

Giulia Sonetti [1,*] , Caterina Barioglio [2] and Daniele Campobenedetto [2]

1    Interuniversity Department of Regional & Urban Studies and Planning, Politecnico di Torino and Università di Torino, Viale Mattioli, 39, 10125 Turin, Italy
2    Department of Architecture & Design, Politecnico di Torino, Viale Mattioli, 39, 10125 Turin, Italy; caterina.barioglio@polito.it (C.B.); daniele.campobenedetto@polito.it (D.C.)
*    Correspondence: giulia.sonetti@polito.it

**Abstract:** Universities play a crucial role in the short-term implementation of education for sustainable development goals (SDGs). The fourth SDG aims to "ensure inclusive and equitable quality education and to promote lifelong learning opportunities for all". Indeed, SDG4 is not intended as a goal in itself, but rather, a tool to achieve different goals and explore the best practices, via deductive-theoretical or inductive-experiential methods. Still, current literature on education for SDGs does not always consider the infrastructural and practical factors affecting the success or the failure of the practices mentioned above. The main purpose of this paper is to organize and describe a set of ongoing education for sustainability strategies that took place from 2016 to 2019 in Italian universities. Eighteen best practices have been collected after a national call by the Italian Network of Sustainable Universities (RUS), that aimed to map the current landscape of SDGs-related actions. Data have been analyzed based on the qualitative description provided by each university, according to four criteria: trigger, course type, approach (top-down/bottom-up) and declared mission. Results are depicted as a map of the current Italian higher education system, where a predominant mission (teaching) and a prevalent driver (top-down) have been found as the frequent features of SDGs educational initiatives. Further developments highlight the value of this first country-wide mapping of the Italian Higher Education Institutions implementing SDGs in their activities, that can avoid the isolation of individual experiences and, most importantly, can suggest some comparability and transferability criteria for similar cases.

**Keywords:** transdisciplinarity; mission-oriented; SDGs implementation; higher education institutions; partnership for the goals

## 1. Introduction

The United Nations' "Transforming Our World: The 2030 Agenda for Sustainable Development" is one of the most ambitious and influential global agreements in recent years. The agenda, with the 17 Sustainable Development Goals (SDGs) at its basis, is a framework to deal with the world's most urgent challenges—including inequalities, climate change and new economic models—in all countries and all people by 2030. While the 2030 Agenda represents an excellent opportunity for the change demanded by the entire society, target 4.7 is specifically related to education for sustainable development (ESD): "By 2030, ensure that all learners acquire the knowledge and skills needed to promote sustainable development, including, among others, through education for sustainable development and sustainable lifestyles, human rights, gender equality, promotion of a culture of peace and non-violence, global citizenship and appreciation of cultural diversity and culture's contribution to sustainable development" [1].

As it happens when sustainable development goals (SDGs) need to be implemented on a local basis, Education for Sustainability (EfS) in practice translates into a complex socio-technical phenomenon. While plural perspectives on EfS are encouraged both by practitioners and researchers, there is also a danger that such pluralism may encourage dominant political ideologies and consolidated corporate power that undermine an ecocentric perspective, or disregard significant differences at the 'grass-roots' level of practice of EfS—both in goals and orientation, as well as the level of educational programs.

In this paper, we intentionally refer to EfS, and not to sustainability education, to highlight the social nature of the concept. Defining education for sustainability, more than "about" and "in", requires an even more deep rethinking on how to change individual attitudes and behaviours toward just social structures and regenerative economies [2–4].

Universities have always been considered as significant contributors to the pursuit of sustainability education initiatives, e.g., [5]. EfS is directed to new generations of leaders and local actors, to contribute to the promotion of sustainability in the socio-technical systems. Higher education institutions (HEIs) often manage large scale portions of cities (e.g., buildings, laboratories, dormitories), in which sustainability principles could be "practised after preached". In this sense, the contribution of HEIs to SDGs implementation goes well beyond the curricula development, and HEIs can be considered as learning communities, in which a variety of practices, discourses and policies coalesce, leading to the elaboration of complex and changing representations and practices of sustainability, and even more difficult to grasp behavioural changes [6–8].

However, HEIs still struggle to implement the new paradigms of EfS into current curricula and operations [9,10]. European HEIs have been quite recently taking the first steps [11,12], integrating sustainable development (SD) into curricula, operational, and research strategies. Such actions have been recognised to be instrumental in providing students with sustainability skills [13] and preparing graduates who are sustainability literate [14], but still, the way to enact the changes is unclear.

The Sustainable Development Solutions Network (SDSN) Australia/Pacific [15], that brings together members in the region to develop and promote solutions, policies and public education for sustainable development, highlights six different ways in which universities can embed sustainable development (SD):

- Including SD into all undergraduate and graduate courses, as well as graduate research training [16,17];
- Delivering training on SD to all curriculum developers, course coordinators and professors [18,19];
- Offering executive education and capacity building courses for external stakeholders based on SD [8,20];
- Defending the implementation of national and public education policies that support education for SD [12,21,22];
- Involving students in the co-creation of learning environments that sustain learning on SD [23,24];
- Developing courses directed to real-world collaborative projects for change [25].

The recent creation of the Italian Network for Sustainable Universities (RUS) [26], recognised by the Conference of Italian University Rectors (CRUI) in July 2015, is aligned with the SDSN aims. It is part of a national institutional resetting on the SDGs implementation, intending to coordinate the actions of all campuses willing to shift the business as a usual model towards a just, sustainable future.

At the time of this paper (July 2019), RUS counts 68 Italian universities (74% of all the Italian universities), and it is continuously growing. RUS is composed of six working groups on different topics: climate change, education, energy, food, mobility and waste, and it collaborates with other sustainability-related national associations, such as AIESEC (Association Internationale des Etudiants en Sciences Economiques et Commerciales) [27] and ASviS (Italian Alliance for Sustainable Development) [28].

The RUS institutional aims are to promote the SDGs framework and the sustainability concept shift toward a holistic pattern in HEIs, both in their practical operations and in their educational offering. To pursue these aims, RUS relies on the single green teams inside each university to overcome

the local infrastructural and practical factors, affecting the success or the failure of the RUS initiatives. One of the RUS' first moves has been the collection of best practices in Education for Sustainability via a national call. The call, harvesting initiatives from 2016 to 2019, aimed to understand, via a semi-structured survey, what strategies have been put in place to implement EfS in the current Italian educational system.

In this paper, we move from the results of this call to describe the state of the art of SDGs' implementation in Italian universities. We presented the analysis of 18 self-selected case studies after a "call for best practice in Sustainability Education" in 2017, across all RUS members. We filtered and described the reported activities according to declared goals and approaches, so that we can read the elements of university governance, curricula, contents and methods toward further integration of sustainability aspects. Universities themselves declared data, but where information was missing, the authors searched on their institutional website for information such as the number of students, main disciplinary foci, type of initiative as declared at the time of the survey, main objectives and the presence of European funds. Then, we grouped them in the kind of macro-areas of institutional change strategies toward SD, as in the methods/approaches found in the literature review. We used an Excel spreadsheet, where we collected, systematised and reorganised recurrent features dividing declared and inferred data, according to specific categories drawing from the literature review and the RUS' survey structure (level, approach, urban outreach, sustainable development goals (SDGs), driver, mission). Results have been systematised in a final map, representing metaphorically the actual structure of the Italian educational system. Within this representation, we aimed to communicate, at a glance, the experiences mapped through the survey for embedding EfS.

This paper's conclusions provide the results of the first mapping exercise of current EfS implementation strategies in Italian HEIs, highlighting the various drivers and challenges among the Italian RUS members that shared their best practice and paved the way for a further positive influence to the national territory.

## 2. Literature Review

In this section, we present the findings of our literature review about sustainability-embedding strategies in higher education institutions.

Evans defines sustainability as "a set of life-ways, lived within specific historical circumstances. Within these life-ways, considerations of the long-term equilibrium of health and integrity remain the central focus for communities" [29]. We start from Evans' point and place sustainability education within the traditions of critical pedagogy, political economy, and globalization, where education for sustainability (EfS) should be the primary focus for suggesting an alternate way of living [19,30].

Education has become a central pole to the achievement of sustainable development goals (SDGs); among these, one stand-alone goal is dedicated to education (SDG4). It is mentioned in targets about other goals, and it is undoubtedly linked to other goals in some way. The agenda covers an extensive set of challenges and, according to [17], the expertise of HEIs is essential for the achievement of the goals; furthermore, SDGs cannot be attained without these institutions.

The literature on how universities are engaging with SDGs implementation is still in an early stage, yet, there has been some research documenting how universities are taking actions to embrace the SDGs within their institutions [31].

The analysis by Aikens et al. [32] combines the quantification of geographic and methodological trends of EfS with qualitative analysis of content-based themes. The majority of articles reviewed were non-empirical; empirical articles overwhelmingly focused on teaching and learning directives, rather than exploring the complexity of policy development or enactment.

A worldwide survey analysing 167 responses has been performed by Filho et al. [16] to collect data on the SDGs and sustainability teaching at universities, mapping them according to the SDNS categories. Most of the respondents claimed to have some knowledge about the SDGs and agreed with their integration at higher education institutions beyond institutional commitment or teaching.

However, the main issue is the lack of concrete and practical integration of the SDGs, since the results from the survey showed much lower levels of application. Some respondents were using the SDGs as key course content, others as a topic addressed in the broader curriculum, others as part of the assessment, but application overall was patchy, despite the opportunity for the SDGs to be used to drive further momentum concerning education for sustainable development.

The research by Lozano et al. [33] provides a holistic analysis on how HEIs have engaged in efforts to embed better environmental and sustainability issues into their system (including institutional framework, education, research, campus operations, outreach and collaboration, on-campus life experiences, and assessment and reporting). A survey answered by 87 respondents from 70 HEIs worldwide revealed that many HEIs have engaged in, and are continuing to participate in, sustainability efforts. However, this research also confirms that, in general, the implementation of sustainability in HEIs has been compartmentalized and not holistically integrated throughout the institutions. The results indicate that there is a strong relation among SD commitment, implementation, and signing of international declarations, but that further research is needed to investigate longitudinal differences in the commitment to and implementation of SD and to explore the differences between the lagging and leading HEIs.

The works by Mulder et al. [23,34] crucially analyse the process of changing engineering universities towards SD, outlining the types of changes needed, concerning approaches, visions, philosophies and cultural change: "instead of adding SD to an unsustainable curriculum, we should rebuild curricula by taking the contribution of a field of expertise to SD as the leading principle for curricula" [20] p. 216. The study Gasca-Pliego et al. [35] highlights what skills connected to SD should be taught to economics students so that they can question economic rationality paradigms, competition without limit, and the conventional focus on self-interest, embracing instead values such as solidarity, cooperation, equality, and mutual respect.

Eisler et al. [36] propose that universities are the "elected" places for the transformation of people and society, enabling young people to acquire the competencies that citizens need to live sustainably, at personal, professional, and community levels.

The SDGs framework may help in enacting such an integrated viewpoint, proceeding with coordinated actions on two tracks: one regards the implementation of the education for sustainability (EfS), stressing its potential to orientate the civic sense; the other, for practising what it is preached in the classrooms, benefitting the transition moment of students enrolling or hiring through new recruits experiencing concrete sustainable practices taking place in daily campus operations [7,8].

The complexity and the interconnection of all those issues, plus the exploratory nature of this first mapping exercise within the Italian context, made us choose qualitative methods to carry out the analysis of current strategies for SDG implementation in the Italian academic institutions.

## 3. Methods

### 3.1. Analytical Framework

The main purpose of this paper is to map what is happening in Italian universities in regard to sustainability implementation. To map these initiatives from a holistic perspective, viewing them as complex, multifaceted elements embedded in a multi-layered network of relations, we adopted a qualitative research method [25,37]. Current literature consists mostly of quantitative analysis on the attempts to incorporate EfS into the curricula, disregarding the social aspect and the context factors affecting its implementation [38–40]. Most of the results of the quantitative method put experiences very far from their local contexts and ecosystem of actors, barriers and drivers. Such data are not able to represent the effective integration of the concepts of EfS into course contents and methods.

We instead adopted a qualitative method aimed at reading, organising and reporting data collected from a national call for best practices in EfS. We described the ongoing strategies in the current educational infrastructure with the method presented below, in three steps.

The primary source of data was the first national call for best practices in sustainability education in Italian Higher Education Institutions (HEIs). It resulted in a report titled "La didattica per lo sviluppo sostenibile negli Atenei italiani—Best practice" (Education for Sustainable Development in Italian Universities—best practices), published in open access [41]. The sample participating in the survey and its design was set and made available by the Italian Network of Sustainable Universities (RUS) in the report found in the same website [42]. The structure of the call was very open. However, for each university, there was a page limit of three and a set of mandatory fields to complete, i.e., the title of the initiative, the description, criticalities and solutions, impacts and expected results.

The call was presented on the 10 July 2017, when RUS organized the first national conference on "Education for sustainable development in Italian universities".

The meeting was held at the Ca' Foscari University of Venice, and it was the first occasion to invite the fifty-one (at that moment) RUS members to present their best practices related to innovative EfS. The objective was twofold: on the one hand, to identify the different approaches and activities already existing on the national territory; on the other hand, to seize the many sensitivities and differences in the local contexts, allowing members to network with others about similar and different approaches.

One limit of this work is, therefore, the partial view on the entire Italian system, since just eighteen universities out of fifty-one voluntarily answered the call. The results cannot track down any unsuccessful attempts, or long-term initiatives of the universities that did not, or could not, participate in the national call.

Another limitation concerns the criteria by which we reported and systematised the data, since the experiences have been submitted using differing approaches to format and content. Some universities presented just one well-detailed best practice, others reported several initiatives or just proposals for future courses or events; the non-homogeneous nature of the results derives from the openness of the survey done by RUS, aiming not at highlighting best or worse performance among members, but rather at collecting the various examples to encourage a wider spread of such initiatives.

Despite these limitations, the results can populate the first map of current principal strategies of EfS in the Italian context. Plus, it contains a synchronic overview of such initiatives, so that the policy framework and the collective actions toward sustainability embedding may cross-fertilise each other.

The second part of our analysis compared the collected experiences to each other. We used an Excel spreadsheet where we collected, systematised and reorganised recurrent features, dividing declared and inferred data according to specific categories (university name, number of students, type, title of the initiative, EU funding, goals and level, approach, urban outreach, sustainable development goals (SDGs), driver, the mission of the initiative, as described in Tables 1 and 2), drawing from the literature review and the RUS' survey structure (Tables 3 and 4).

In the third step of our method, after analysing the results (Figures 1–3) we drew an actual map as a synoptic graphic representation, to show all the initiatives resulting from the call. We departed from the state of the art of Italian HEIs (Figure 4) to interpret and systematize which actions, loci of change and drivers the 18 universities included in the current Italian higher education system. The map (Figure 5) of the Italian urban territory shows streets that stand for the curricular paths, temples that are the compulsory modules, tents which symbolise pedagogical experiments run apart from university environments, and other urban elements representing elements of the Italian educational infrastructure, in an attempt to depict the peculiarity of the Italian setting.

The following data collection paragraph describes how we carried out the data collection and presents the data reported by the 18 universities of the sample.

### 3.2. Data Collection

The first part of the data collection describes the characteristics of the eighteen universities of the self-selected sample. Universities themselves reported the data, but where information was missing, the authors searched on their institutional websites for information, such as the number of students, main disciplinary foci, type of initiative as declared at the time of the survey, main objectives and the

presence of European funding for the initiative. The tables below summarize and systematize the information extracted from the calls of each university; the categories are defined in Table 1 and the data are reported in Table 3.

We organized the results according to the categories explained in Table 2, inspired by the works on the quantitative scanning of SDGs implementation in HEIs, as cited in the literature review section.

This data collection provides the elements we represented as roads, houses and urban landmarks, in the map of the current EfS actions in Italian universities.

We chose this conceptual representation to let the actions on the current HEIs' structure emerge and to communicate them clearly to a non-expert audience. Instead of defining what universities "should" do, we conducted a qualitative and comparative reconnaissance, to explore what is happening in the actual system of HEIs, which metaphorically form a city (Figure 5).

We did not consider "ex-novo" initiatives, so that this research may constitute a collection of hints for future actions about impacts and feasibility options per each regional context.

**Table 1.** Categories by which declared data from the first national call on "Education for sustainable development in Italian universities" have been organized.

| Category | Definition |
| --- | --- |
| University name | Name of the HEI that is carrying on the initiative, as declared in the RUS survey. |
| N. Students | The number of students at the university (to account the size of the HEI). |
| Type | Type of the HEI in which the initiative is carried on. SSH = institutions devoted primarily to Social Science and Humanities. STEM = institutions devoted primarily to Science, Technology, Engineering and Mathematics. Arts = institutions devoted primarily to arts. |
| Title | Title of the initiative as declared in the RUS survey. |
| EU funding | Presence of EU funding for the initiative as declared in the RUS survey. |
| Goals | Main goals of the initiative as declared in the RUS survey. |

**Table 2.** Categories by which interpreted data from the first national call on "Education for sustainable development in Italian universities" have been organized.

| Category | Meaning |
| --- | --- |
| Level | Loci of the initiative in the educational management structure in which the educational activity happens. It could be a one-time and not a repeated action; it may be a university course explicitly dedicated to sustainable matters, or a program (an entire sequence of classes), or a whole university unit (a department, a green team, UNESCO chair, etc.), or a virtual place where the educational activities are conducted by a network of researchers or practitioners. |
| Approach | The pedagogical method applied in the observed case study. i.e., frontal lecture, experiential learning, problem-project based learning, etc. A further definition of all the approach categories is given in Table 5. |
| Urban Outreach | The stakeholders involved in the educational experience, i.e., NGOs, city councils, social welfare associations, etc. |
| Sustainable Development Goals (SDGs) | The SDGs embedded and addressed through that educational practice, even when not elicited. For instance, a leadership training education course aims at a quality of education for all (SDG4). It targets stakeholders of industries and decision-makers to take action in the responsible production and consumption patterns, with developed countries taking the lead (SDG12). |
| Driver | The direction of the process for initiating and developing the educational activity: it could be top-down when decided from the leadership or the university authorities, or bottom-up, when born among student associations or spontaneous university staff uniting towards a sustainability goal. |

**Table 2.** *Cont.*

| Category | Meaning |
|---|---|
| Mission | The type of task (research, teaching, and third mission). For instance, students' engagement activities in energy conservation projects fall into the research and teaching mission, while the inclusion of citizens in a waste-collection programme around the neighbourhood recap the third mission. We define the third mission as the effort to link the university's activity with its socio-economic context, according to [43]. |
| On the map | This column refers to 12 types of action to implement HEI that have been performed by Italian University that participated in the RUS call. These types have been identified by the authors and are described in detail in Section 5. |

The second part of the data collection summarizes the qualitative analysis started from the RUS survey and then interpreting the description of the case study, as reported in the survey.

**Table 3.** The declared data by Italian Universities, self-selected after a "call for best practice in Sustainability Education" in 2017, made by the "Italian Network for Sustainable Universities" (RUS).

| University | N. Students (2018) | Type | Title | EU Funding Integration | Goals |
|---|---|---|---|---|---|
| Università di Bologna | 84,720 | SSH + STEM | UniBo Green Office | Yes | Student engagement; networking among universities; networking among urban stakeholders; job market skills. |
| Politecnico di Milano | 45,000 | STEM | Polimi4SDGs | No | Mapping of SDGs related activities; data collection about education al activities referred to Agenda 2030 goals. |
| Politecnico di Milano | 45,000 | STEM | 4 MSc programs | No | Theoretical framework of environmental engineering, environmental sustainability; energy for development and sustainable architecture. |
| Politecnico di Milano | 45,000 | STEM | Postgraduate courses | No | Theoretical Framework; specialization on specific topics (Energy, Buildings, infrastructures, temporary reuse, renewal energies). |
| Politecnico di Milano | 45,000 | STEM | Honorous path: engineering for sustainable development | No | Specialization in engineering for sustainable development. |
| Politecnico di Milano | 45,000 | STEM | Unesco chair: energy for sustainable development | Yes [UN] | Specialization in energy for sustainable development. |
| Politecnico di Milano | 45,000 | STEM | Postgraduate: Coopera(c)tion: knowledge and skills for sustainable cities in the global South | Yes | Social Impact; soft skills; sustainability awareness. |
| Politecnico di Milano | 45,000 | STEM | 2 EU Pr: LeNSin + SUSTAIN T | Yes | Networking among universities; improve internationalization; intercultural cross fertilization, accessibility of higher education. |
| Politecnico di Milano | 45,000 | STEM | 2 MOOCS on sustainability | No | Specialization in sustainable building design; social entrepreneurship. |
| Politecnico di Torino | 31,500 | STEM | 2 MSc programs: systemic design and sustainable architecture | No | Specialization in sustainable architecture; systemic design. |
| Politecnico di Torino | 31,500 | STEM | Green Team | No | Resources optimization; integrated environmental education for university students; Participation in university governance; developing strategies for disseminating among university students environmental responsiveness and sustainability culture. |

**Table 3.** *Cont.*

| University | N. Students (2018) | Type | Title | EU Funding Integration | Goals |
|---|---|---|---|---|---|
| Politecnico di Torino | 31,500 | STEM | Honorous path: Young Talent Program | No | System thinking; complexity awareness; social impact; local solutions; soft skills. |
| Politecnico di Torino | 31,500 | STEM | SDGs mapping | No | Mapping of SDGs related activities; data collection about educational activities referred to Agenda 2030 goals. |
| Università Ca' Foscari | 21,529 | SSH | Active Learning Lab—Urban Innovation | No | Network among urban stakeholders; knowledge transfer; innovation hub; job market skills; social impact. |
| Università dell'insubria | 10,000 | SSH | Waste Management Feasibility Project | No | Waste management; environmental awareness. |
| Università di Bari | 48,000 | SSH | Environmental sustainability | No | Social impact; soft skills; system thinking. |
| Università di Napoli "L'Orientale" | 11,685 | SSH | Project within "Ethics and Market" course | No | Community needs assessment; local set of solutions; sustainability awareness. |
| Università di Napoli "l'Orientale" | 11,685 | SSH | Migrations and sustainable development project | No | Social impact; soft skills; sustainability awareness; improve internationalization; intercultural cross fertilization, accessibility to higher education. |
| Università di Napoli "l'Orientale" | 11,685 | SSH | Open Doors Summer School on Migration Sea Borders Control and Human Rights (CeMiRiMed) | Yes | Networking among universities; policies; knowledge transfer; system thinking. |
| Università di Parma | 22,500 | SSH + STEM | Italian Center for Environmetal Research and Education | / | Transdisciplinary research and education. |
| Università di Parma | 22,500 | SSH + STEM | Department of Chemical Life Sciences and Environmental Sustainability | / | Transdisciplinary research and education. |
| Università di Parma | 22,500 | SSH + STEM | BSc Food System: Sustainability Management and Technology | / | Social impact; system thinking. |
| Università di Parma | 22,500 | SSH + STEM | Sustainability in University Teaching Programmes | No | Sustainability education; SSH integration; network among urban stakeholders. |
| Università di Perugia | 23,877 | SSH + STEM | MSc in Circular design | No | System thinking; sustainability education; complexity awareness; job market skills orientation. |
| Università di Siena | 16,400 | SSH + STEM | Sustainability open course (6 CFU) | No | Sustainability theoretical framework; external stakeholders engagement. |
| Università di Siena | 16,400 | SSH + STEM | Summer school for Sustainable Development | / | Sustainability theoretical framework; external stakeholders engagement. |
| Università di Torino | 70,500 | SSH | Unito Go | / | Students' engagement; network among universities; network among urban stakeholders; job market skills. |
| Università di Torino | 70,500 | SSH | Leadership Training for Education for Sustainable Development | No | Sustainability theoretical framework; soft skills; local set of solution. |
| Università di Torino | 70,500 | SSH | Postgraduate program in Socio-environmental sustainability of Agro-food network | No | Job market skills; sustainability theoretical framework; complexity awareness. |
| Università di Torino | 70,500 | SSH | Unito for International Cooperation | Yes (?) | Social impact; network; sustainability awareness; complexity awareness. |
| Università IUAV | 4600 | ARTS | No Title—Trigeneration powerplant and organization of visits for students | No | Sustainable energy production; sustainability awareness. |

**Table 4.** The categorization of qualitative data reported by the 18 Italian universities, self-selected after a "call for best practice in Sustainability Education" in 2017, held by the "Italian Network for Sustainable Universities" (RUS). Source: authors' elaboration.

| University | Title | Level | Approach | Urban Outreach | SDGs | Driver | Mission | On the map | Legend |
|---|---|---|---|---|---|---|---|---|---|
| Università di Bologna | UniBo Green Office | University unit | Experiential | City Council | 11; 17; 4; 13; 12; 6; 8 | Bottom-Up | 3rd mission | Renew a part or an entire existing building | 12 |
| Politecnico di Milano | Polimi4SDGs | University unit | / | / | / | Top-Down | 3rd mission | Draw a map, give a compass | 1; 2 |
| Politecnico di Milano | 4 MSc programs | Program | Problem/project-based; lectures | / | 7; 9 | Top-Down | Teaching | Building something new | 11 |
| Politecnico di Milano | Postgraduate courses | Program | Problem/project-based; lectures | / | 7; 11; 9 | Top-Down | Teaching | Building something new or do building maintenance | 11; 5 |
| Politecnico di Milano | Honorous Path: engineering for sustainable development | Course | Problem/project-based; lectures | / | 7; 9 | Top-Down | Teaching | Building something new or do building maintenance | 11; 5 |
| Politecnico di Milano | Unesco chair: Energy for sustainable development | University unit | Problem/project-based; lectures | / | 11; 7; 13; 16; 4 | Top-Down | Teaching; Research | Elevate temples | 9 |
| Politecnico di Milano | Postgraduate: Coopera(c)tion: knowledge and skills for sustainable cities in the global South | Program | Problem/project-based; lectures | / | 10; 4 | Top-Down | Teaching; 3rd Mission | Put up tents outside | 4 |
| Politecnico di Milano | 2 EU Pr: LeNSin + SUSTAIN T | Network | Experiential; challenge-based; | NGOs; Local Health Offices; Social Welfare Associations | 4; 5; 16; 8; 10 | Top-Down | Teaching | Trace trails | 7 |
| Politecnico di Milano | 2 MOOCS on sustainability | Course | Online lectures | / | 4; 11; 7; 5 | Top-Down | Teaching | Put aerial outside the houses | 8 |
| Politecnico di Torino | 2 MSc program: Systemic design + Sustainable architecture | Program | Problem/project-based; lectures | / | 7; 9; 11 | Top-Down | Teaching | Building something new do building maintenance | 11; 5 |
| Politecnico di Torino | Green Team | University unit | Holistic | Local experts, Municipal Council; NGOs; Social Welfare Associations | 11; 17; 4; 13; 12; 6; 8 | Top-Down | Operational aspects; 3rd mission | Make it rain | 3 |

**Table 4.** *Cont.*

| University | Title | Level | Approach | Urban Outreach | SDGs | Driver | Mission | On the map | Legend |
|---|---|---|---|---|---|---|---|---|---|
| Politecnico di Torino | Honorous Path: Young Talent Program | Course | Challenge-based; problem/project-based; experiential; transdisciplinarity | Local Experts; Social Welfare associations, business | 11; 7; 4; 7; 9; 10; 12; 13 | Top-Down | Teaching | Building something new | 11 |
| Politecnico di Torino | SDGs mapping | University unit | / | / | / | Top-Down | 3rd mission | Draw a map, give a compass | 1; 2 |
| Università Ca' Foscari | Active Learning Lab—Urban Innovation | Course | Lectures | Business; City Council; NGOs | 8; 4; 1; 11; 12; 17 | Top-Down | Teaching; 3rd Mission | Do building and soil maintenance | 5 |
| Università dell'insubria | Waste Management Feasibility Project | University unit | Experiential | / | 12 | Top-Down | Operational aspects | Trace trails | 7 |
| Università di Bari | Environmental sustainability | Course | Challenge-based; problem/project-based; experiential | / | 4; 12; 17; 13 | Top-Down | Teaching | Do building and soil maintenance | 5 |
| Università di Napoli "L'Orientale" | Project within "Ethics and Market" course | Spot initiative | Problem/project-based; experiential | / | 3; 4; 10; 11; 12 | Top-Down | Teaching; Operational aspects | Make it rain | 3 |
| Università di Napoli "l'Orientale" | Migrations and sustainable development | Spot initiative | Challenge-based; experiential | NGOs; Local Health Offices; Social Welfare Associations | 10; 4 | Top-Down | Teaching; 3rd Mission | Put up tents outside | 4 |
| Università di Napoli "l'Orientale" | Open Doors Summer School on Migration Sea Borders Control and Human Rights (CeMiRiMed) | Spot initiative | Challenge-based | NGOs (local and international); | 4; 5; 16; 8; 10 | Top-Down | Teaching; 3rd Mission | Building playfields for sustainability education | 10 |
| Università di Parma | Italian Center for Environmental Research and Education | University unit | Lectures | Schools | 4; 17 | Top-Down | Teaching; Research | Power factories and labs with SDGs fuel | 6 |
| Università di Parma | Department of Chemical Life Sciences and Environmental Sustainability | University unit | Lectures | Business | 4; 17; 12 | Top-Down | Teaching; Research | Power factories and labs with SDGs fuel | 6 |
| Università di Parma | BSc Food System: Sustainability Management and Technology | Program | Lectures; experiential | Business | 4; 17; 12; 1 | Top-Down | Teaching | Houses along the path | 5; 11 |

**Table 4.** *Cont.*

| University | Title | Level | Approach | Urban Outreach | SDGs | Driver | Mission | On the map | Legend |
|---|---|---|---|---|---|---|---|---|---|
| Università di Parma | Sustainability in University Teaching Programmes | Spot initiative | Interdisciplinarity; staff, professor and student engagement | Local experts | 17; 16 | Top-Down | Teaching | Rain collector tank | 3 |
| Università di Perugia | MSc in Circular design | Program | Interdisciplinarity; problem/project-based courses; local stakeholder engagement | Business | 8; 9; 12 | Top-Down | Teaching; Technological Transfer | Building something new do building maintenance | 5; 11 |
| Università di Siena | Sustainability open course | Course | Lectures | Local experts | 4 | Top-Down | Teaching | Put aerial outside the houses | 8 |
| Università di Siena | Summer school for Sustainable Development | Spot initiative | Interdisciplinarity; problem/project-based courses; local stakeholder engagement | Local experts, Municipal Council; NGOs; Social Welfare Associations; business | 4; 17 | Top-Down | Teaching | Building playfields for sustainability education | 10 |
| Università di Torino | Unito Go | University unit | Experiential | City Council | 11; 17; 4; 13; 12; 6; 8 | Bottom-Up | 3rd mission | Renew a part or an entire existing building | 12 |
| Università di Torino | Leadership Training for Education for Sustainable Development | Spot initiative | Problem/project-based courses; multidisciplinarity | / | 12; 4 | Top-Down | Teaching; Operational aspects | Do building and soil maintenance | 5 |
| Università di Torino | Postraduate program in Socio-environmental sustainability of Agro-food network | Program | Problem/project-based courses; multidisciplinarity | Local experts, Municipal Council; NGOs; Social Welfare Associations | 4; 2; 12; 15; 17 | Top-Down | Teaching | Do building and soil maintenance | 5 |
| Università di Torino | UniTo for International Cooperation | University unit | Experiential; problem/project-based courses; multidisciplinarity | NGOs; Local experts; Municipal Council; Businesses; | 1; 4; 5; 6; 10; 11; 12; 13; 14; 15; 16; 17 | Top-Down | Teaching; 3rd Mission | Trace trails and bridges | 7 |
| Università IUAV | No Title—Trigeneration powerplant and organization of visits for students | University unit | Dissemination | Business | 7 | Top-Down | Operational aspects | Trace trails and bridges | 7 |

## 4. Results

Starting from the systematization of collected data, we drew on preliminary insight to cluster-specific strategies for embedding EfS into current Italian HEIs.

The following charts compare, at a glance, the universities' profiles and locations. Most of the Italian universities that joined the survey are located in northern Italy, where the Ministry of Education and the system of regional industries and international private partnerships allocate more funding (Figure 1 right).

Science, technology, engineering and mathematics institutions (STEM) and social science and humanities-focused universities (SSH) are almost equally implementing sustainability strategies. This last finding suggests that teaching sustainability could be set as a goal regardless of the teaching focus of the institution (Figure 1 left).

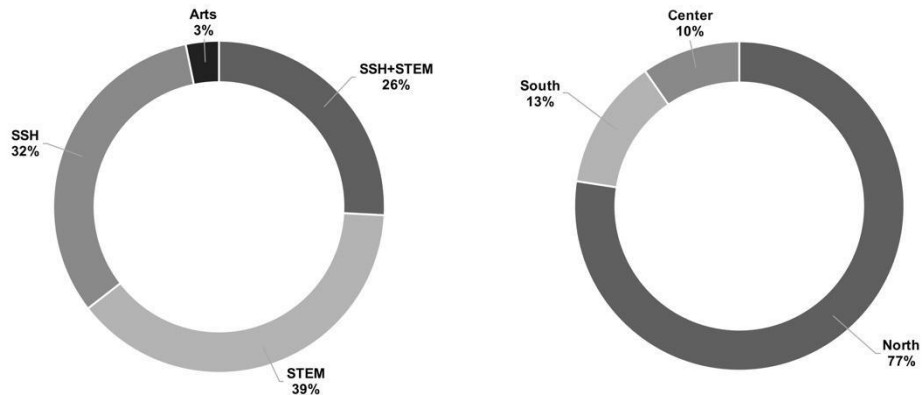

**Figure 1.** Profiles of the universities engaged in the study. **Left**: Disciplinary focus is broken down for Social Science and Humanities (SSH); Arts; Science, Technology, Engineering and Mathematics (STEM). **Right**: Geographical distributions within the Italian territory. Source: authors' elaboration.

Specific drivers are top-down actions proposed by the university government (85% of the cases), followed by experiential training for students (7%), and bottom-up initiatives (6%) (Figure 2 left). It is worth noting that the "experiential training" initiatives, meaning those carried out in NGOs or Erasmus Plus projects, were not initially meant to be mapped as EfS actions. Nevertheless, the experiential training programmes, mandatory for those enrolled in a university course, could be essential to helping students develop knowledge, skills, and values from direct experiences outside of a traditional academic setting. These experiences include internships, service learning, undergraduate research, study abroad, and other creative and professional work experiences. Well-planned supervised and assessed experiential learning programs can stimulate EfS by promoting interdisciplinary learning, civic engagement, "green" career development, cultural awareness, leadership, and other professional skills. Top-down initiatives can be found at different levels of universities' structures: most frequently, they happen within a dedicated university unit, which characterizes more than one/third of all the cases. One-time initiatives, i.e., single initiatives that do not repeat over time, dedicated EfS courses, or entire programs (postgraduate and undergraduate) occur in about one/fifth of the cases. On the other hand, initiatives taking place among collaborating university networks are still significantly less frequent at the time of the survey (Figure 2 right).

Almost two-thirds of the missions of the reported initiatives fall into the teaching area, followed by those into outreach activities that seek to generate knowledge outside academic environments to focus on social, cultural and economic development. Fewer initiatives include operational aspects (13%) and research activities (5%) (Figure 3 left).

The pedagogical approaches and aims of these initiatives vary a great deal. The typologies by which we gather them are explained in Table 2 and represented in the Figure 3 right. The university respondents did not self-identify these approaches/categories, neither did the survey contain a list to

choose from. The description field was a free text one where we then applied the categories, where we found an appropriate definition or an explicit reference. Methods and tools ranged from lectures (11%) to inter/trans/multi-disciplinary projects (15%). The most frequently represented cases are those involving students, often connecting them to a real-world experience. Experiential learning represents one-quarter of the total approaches; challenge-based ones are one-fifth, and problem/project-based courses count for another one-fifth.

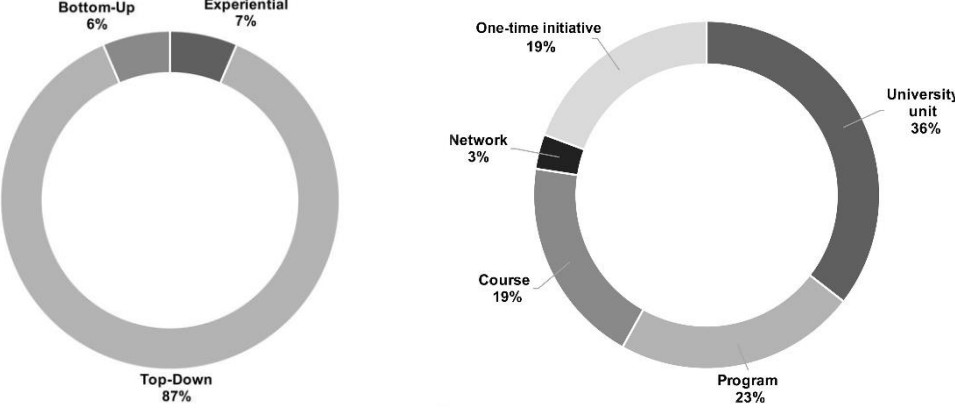

**Figure 2.** The drivers (**left**) and the levels (**right**) at which the initiatives for education for sustainability (EfS) took place within the universities in the sample. Source: authors' elaboration.

The results of the call for best practices may give some insights, referring to a set of aggregated data from the annual RUS survey of 2017. When asked about the presence of a sustainability cross-reference in the university statute, just 29.27% answered affirmatively. A rector's delegate on sustainability issues is present in 68.29% of the cases, while an organizational unit dedicated to sustainability in the university is found in 39.02% of the responses. These aggregated data confirm the lack of a national homogeneous and collective strategy for SDGs' embedding in university curricula.

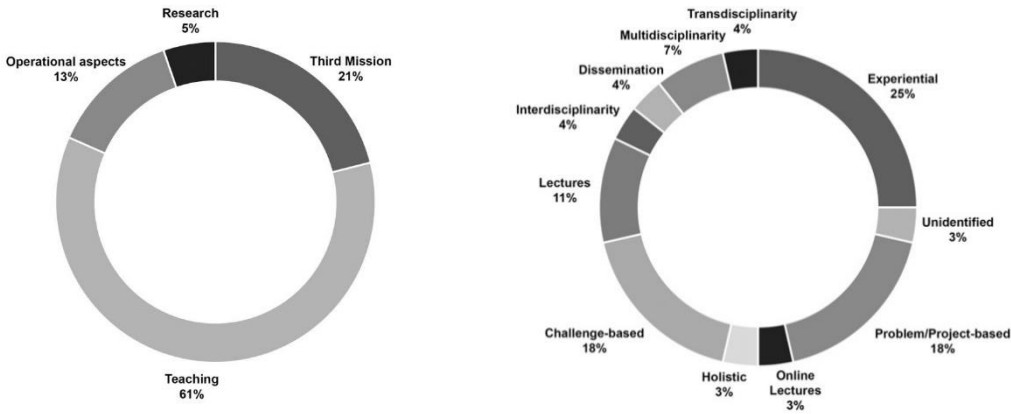

**Figure 3.** The mission (**left**) and the type of approach (**right**) of the reported initiatives within the universities in our sample. If an initiative had multiple missions/approaches, it has been counted twice, so that in the charts, the 100% does not represent all the cases, but only the recurrence of such features)—source: authors' elaboration.

**Table 5.** Definition of the approaches we used to categorize the survey results as in Figure 3 (right), source: authors' elaboration.

| Methods and Tools | Definitions and Sources |
|---|---|
| Transdisciplinarity | Transdisciplinary projects are those in which efforts conducted by persons from different disciplines and backgrounds working jointly to create new conceptual, theoretical, methodological, and translational innovations that integrate and move beyond discipline-specific approaches to address a common problem. Trans-disciplinary work moves beyond the bridging of divides within academia to engage directly with the production and use of knowledge outside of the academy [25,44]. |
| Multidisciplinarity | Multidisciplinarity involves studying a topic using several different discipline perspectives at the same time. Any topic will ultimately be enriched by the sum of the disciplinary perspectives when the multidisciplinary approach overflows disciplinary boundaries. Participants, however, work within the respective frameworks of their disciplines [45–47]. |
| Interdisciplinarity | Interdisciplinarity happens in any study or group of studies undertaken by scholars from two or more distinct scientific disciplines. It is based upon a conceptual model that links or integrates theoretical frameworks from those disciplines, uses study design and methodology that is not limited to any one field, and requires the use of perspectives and skills of the involved disciplines throughout multiple phases of the research process [48]. |
| Dissemination | All the university's activities meant to spread information, knowledge, and opinions widely about the SDGs. |
| Lectures/Online lectures | Any educational talk to an audience, especially one offered to the students in a university. |
| Challenge-based | A "challenge-based" approach is a combined set of relevant challenges approached with a collaborative inquiry. It begins with a content-relevant challenge and is followed by a request for learners to generate their initial thoughts about the challenge, access to student-controlled audio and video resources (essentially mini-lectures) designed to deepen learners' initial thoughts, a chance for small-group discussions about the challenge and resources, and finally a large-group discussion that includes key ideas students have learned and further questions for the instructor [49]. |
| Holistic | The aim and focus of holistic learning are making connections, e.g., connections between subjects or between thinking and intuition [50]. The transdisciplinary approach of holistic type, that overcomes the disciplinary fragmentation, reports a vision of the world and life, as comprehensive as possible, and looks at the human nature with all its complexity and diverse forms of manifestation [51]. |
| Problem/project-based | Problem-Based Learning is a teaching method in which complex real-world problems are used as the vehicle to promote student learning of concepts and principles as opposed to direct presentation of facts and concepts. In addition to course content, it can promote the development of critical thinking skills, problem-solving abilities, and communication skills. It can also provide opportunities for working in groups, finding and evaluating research materials, and life-long learning [52]. While with challenge-based learning students are asked to develop solutions to a complex problem incorporating technology into the process and to propose real-world solutions, the goal of problem/project-based learning is to complete a critical thinking exercise and come up with a project that may solve a specific problem. |
| Experiential training | Experiential training helps students develop knowledge, skills, and values from direct experiences outside of a traditional academic setting. It encompasses internships, service learning, undergraduate research, study abroad, and other creative and professional work experiences [53]. |
| Unidentified | We did not assign categories to those initiatives which were poorly described. |

## 5. Discussion

In this section, we sum up the analysis of best practice in education for sustainability, as reported by the self-selected sample of 18 Italian university members of the RUS. For easier reading, we report the discussion also in a graphic representation. We metaphorically draw the current structure of higher education in the Italian system like a city (Figure 4), where streets are the undergraduate/graduate programmes. Buildings of various types (big/small houses, industries, blocks, temples) are connected by roads and represent the training and educational experiences that students have to pass through. The natural elements help to identify the relationship among these experiences (for example, mountains represent a clear separation between two academic programmes, while other programmes are represented in flat terrain, symbolizing an easier possibility of exchange through a greenfield). This representation does not refer to a particular set of programmes promoted by a specific athenaeum, rather, it collects the most common characteristics of the Italian programmes. The representation is thus to be considered qualitative

rather than quantitative: each object represent a type of activity and the number of objects/experiences represent their uniqueness (one object) or repetition (more than one object) within a typical academic programme. We then place, in such a map, the initiatives related to EfS as they happen in various aspects and functions at the Italian universities, according to the RUS survey (Figure 5). Finally, we drew the 12 main actions (Figure 6) undertaken by the 18 universities in our sample, summarised in a list that may serve for paving the way to a shift for the university of the XXI century.

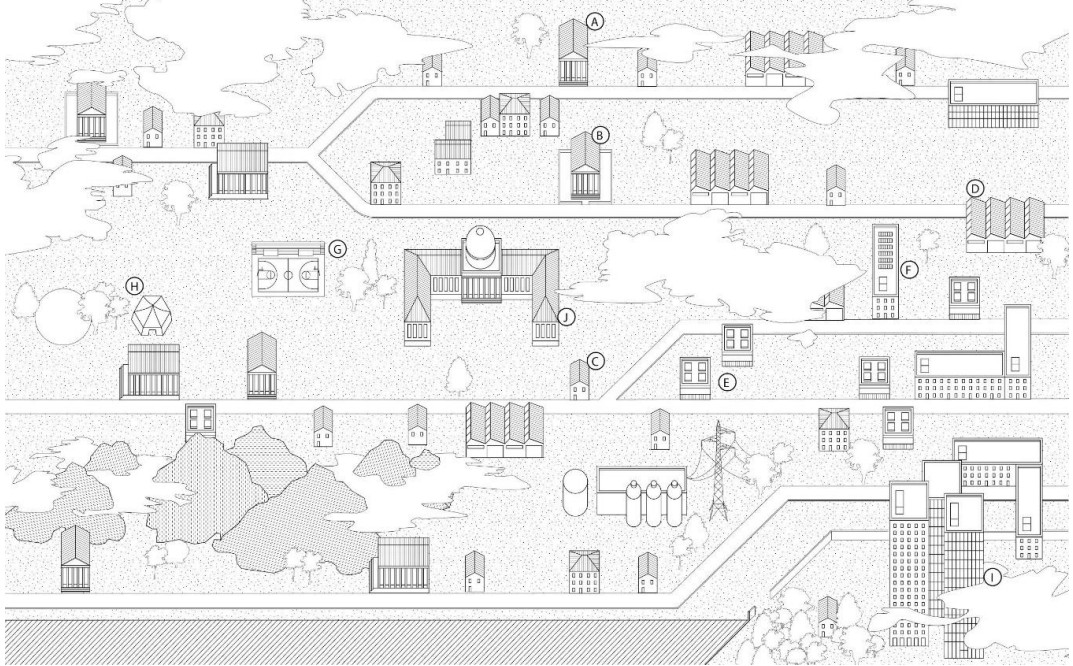

**Figure 4.** The map of the current Italian universities structure. Source: authors' elaboration.

The elements of the map we draw to describe the current Italian universities structure are the following:

A. the temples: the theoretical foundations of disciplines, available to all students (e.g., Physics, Mathematics, History of Architecture, etc.);

B. the temples with fences: the theoretical foundations of disciplines, but specifically dedicated to a certain degree (Fluid Dynamics, Compositional principles, Anatomy, etc.);

C. the houses: a theoretical course inside one or more degree courses (technical physics, structural engineering, interior design, etc.);

D. the factory: a laboratory for experiential learning (wind tunnel, chemistry experiments, architectural model crafting, etc.);

E. the atelier: a course in which a project is a central part (architectural design, electric circuit design, model design, etc.);

F. the building block: the research experience inside a degree course (thesis project, essays, special seminars);

G. the playfield: a limited learning experience around a specific topic (summer school, student challenges, hackathons, etc.);

H. the tent: temporary learning experiences (bottom-up initiatives by students, teamwork around a societal challenge, etc.);

I. the suburbs: outside learning experience from different stakeholders (an internship in companies, in public administrations, in NGOs, etc.);

J. the main building: the administration and general direction of the university.

From a RUS call, we understand that sustainability education in Italian universities is happening, and could happen, in 12 ways (Table 2). We reorganised data presented in Section 4, describing in detail the 12 existing ways to operate HEI implementation within Italian curricula though a new map (Figure 5). We chose this iconic representation to let the sewing, mending and fixing actions on the actual structure emerge and be clear to the variety of stakeholders that could benefit from this mapping exercise. Instead of defining what universities "should" do, we simply represent through a qualitative and comparative picture what is happening in the actual structure of HEIs, that is, the "city" in black in the background of the map. In the representation, each object stands for a type of activity and the number of objects/experiences represent their uniqueness (one object), or repetition (more than one object) within a typical academic programme. Therefore, the spatialization of each action (i.e., the rain above one specific building or the presence of a bridge between two central buildings) is not meant to symbolize anything: the elements in the map are just items depicting a characteristic of the sample we analysed, but their numerosity or position in the map does not have a mirror in the quantitative data list.

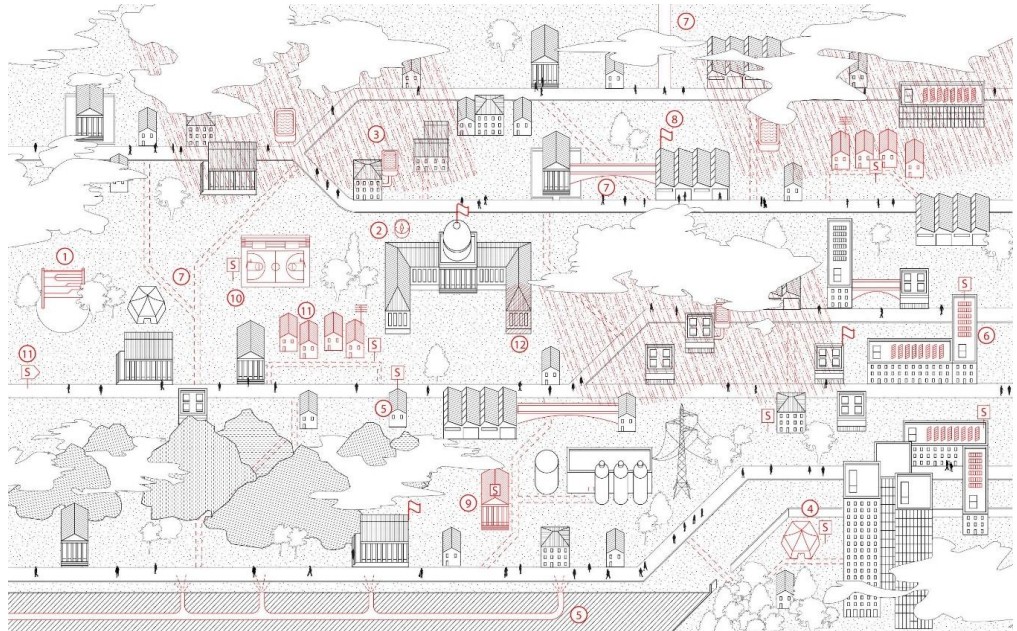

**Figure 5.** The map of a potential university structure holding together sustainability education actions and their interaction within the entire system—source: authors' elaboration.

Here follows the list of the 12 main types of initiatives, as reported in the survey results and in our analysis (Figure 6):

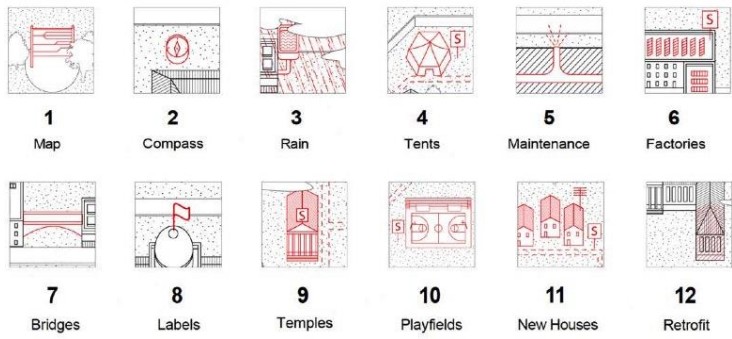

**Figure 6.** The elements of Education for Sustainability, as embedded in the current Italian University System. Source: authors' elaboration.

1—A map: a template by each governing body on how to map their current sustainability implementations, ensuring everyone on campus knows what the institutional goals are and why their efforts are essential to them. According to our data set, Bologna university released its first Sustainable Development Goals (SDGs) mapping report in 2018, to have an understanding of SDGs implementation in its educational, research and third mission offers. Politecnico di Milano promoted the "Polimi for SGDs" initiative in 2017. The Politecnico di Milano is proceeding with the mapping of its internal competencies relevant to the 17 Sustainable Development Goals. The objective is to gather information on how and where the university is responding to the challenge launched by the UN. The Politecnico di Torino, too, is proceeding with the SDGs mapping of its curricula and research products, via a machine-learning algorithm and a human cross-check of results.

2—A compass: a guide for SDGs mapping, to understand which steps can be undertaken for embedding the SDGs in each curriculum. So far, Politecnico di Milano, Politecnico di Torino and Alma Mater Bologna are examples of universities that engaged in a systematic process for SDGs mapping in education, research and third mission activities. A dedicated programme for enhancing SDGs awareness into the university community and the designation of a specific task within the sustainability office have been some of the strategies for encouraging university authorities to address environmental and social challenges and build capacity and ownership of the SDGs [12,21,22].

3—Rain: a top-down strategy to embed SDGs-related questions into most class' assignments, discussions, lectures, case studies, practice-based learning, etc.

The ILO/UN training centre [54] in Italy is successfully carrying on radical practices on sustainability education, offering ecology-centred masters programmes and short courses. Some "rains" are happening in separate ways, with seminars open to everybody enrolled at universities, like the Siena course on Sustainability Literacy [16,17].

4—Tents: special courses to allow students to explore as independently as possible SDGs learning possibilities as training courses or extension projects. They can also be recognised as European Credit Transfer and Accumulation System (ECTS), accounting for the final degree. Examples of this are found in the Neapolitan University "L'Orientale", where students enrol in Erasmus Plus cooperation actions in the third world (migrations and sustainable development) as part of their curriculum. Similarly, in the Politecnico di Milano, a postgraduate course is dedicated to "Coopera(c)tion: knowledge and skills for sustainable cities in the global south". The "Active Learning Lab-Urban Innovation" of the University of Venice, or the executive education course offered to managers and sustainability practitioners by the University of Turin, are examples of university injecting expertise coming from outside.

5—Maintenance: The creation of universities' green teams like in the ones in Turin, Milan, and Venice cases, demonstrate to students the values of practical actions in campus operations related to the physical dimension of sustainability, implemented with the "maintenance" operational aspect in the very physical realm of any university setting.

6—Factories: SDGs can frame research priorities and impact evaluation, to foster the foundation of interdepartmental centres or research groups devoted to the achievements of societal challenges and related SDGs. This happened at Politecnico di Torino and Politecnico di Milano, as an efficient strategy to catalyse existing energies around new SD issues. Parma University created an entire department (of "Chemical Life Sciences and Environmental Sustainability"), under the direction of specific sustainability goals [8,20].

7—Bridges: partnerships to advance the SDGs awareness, mutual reinforcement, implementation and synergic actions. A path may connect universities' plants and facility management offices directly to students' courses or thesis projects. The University of Insubria project on student inclusion in the waste management feasibility plan or the Venice case of the tri-generation plant inside the campus saw students and professors involved in its construction and use. A similar path connects students' internships with living lab offices or green teams, like in the University of Bologna, Venice or Politecnico di Torino. International cooperation actions like the ones undertaken by the University of Turin and

the Politecnico di Milano are examples of roads heading outside the map, and that build a precious connection to the real-world educational field [25].

8—Labels: report on efforts and impacts about SDGs, celebrate success, and to foster grass-roots initiatives and competitions around SDGs implementation. This can be done methodically at the individual, classroom, teacher, course, curriculum, governance and institutional levels, to communicate decision processes, strategic choices and minor improvements at operational and building levels. MOOCs as the one by the university of Siena or the Politecnico di Milano about SDGs literacy is a precious aerial to disseminate sustainability action also outside the map borders [18,19].

9—Temples: Unesco Chairs for Sustainable Development are based in 20 Italian universities, aiming to promote sustainability direction to different education and research fields (including energy, cultural heritage, urban culture, etc.) [55].

10—Playfields: the cultivation of generative social fields, of relationships among learners, educators, parents, community members, and nature, is a powerful gateway to the deeper sources of knowledge. Existing playfields in the Italian university cases are at the summer schools, like the one focused on sustainable development held in Siena, and organised by ASviS in collaboration with the University of Siena—Santa Chiara Lab, Enel Foundation, Leonardo, Italian Network of universities for the Sustainable Development (RUS), Sustainable Development Solutions Network Italia, and the Sustainable Development Solutions—Mediterranean Network. The teachings have concerned, among other things, sectoral policies (public sector, institutions, networks international organizations), science and innovation (agriculture, new materials, architects-engineering and engineering), and the development of private models (b-corp, sustainable finance, new business models). Moreover, the Neapolitan "Open Doors Summer School on Migration Sea Borders Control and Human Rights (CeMiRiMed)", or the Parma, Torino, Bologna working groups on sustainability awareness actions, are good examples of sustainability playgrounds [56].

11—New houses: honours programmes, like the ASP—Alta Scuola Politecnica (High Polytechnic School) course, offered as a joint venture with Politecnico di Torino and Politecnico di Milano, are dedicating two years of extra classes on engineering for sustainable development. Other buildings are directly connecting existing and new houses, giving a direction towards sustainability education, like in Bologna University's Master's in Sustainable Design [57–59].

12—Retrofit: the creation of dedicated unit inside the university organizations, such as a green team, or the empowerment of a students' association, as it occurs in most of the 18 cases, assures the visibility of sustainability intentions, both constituting a reference point for other students and researchers engaged in SDGs, and for external stakeholders willing to collaborate with academia on sustainability topics [23,24].

This mapping exercise demonstrated some structural weaknesses of the Italian setting, where a truly holistic effort toward a systemic sustainability shift of the entire educational system is not yet occurring.

The central gap we see related to the international literature on how to embed education for sustainability in current higher education institutions' structures and infrastructures is the lack of an inter/trans-disciplinary literacy as the preliminary ground to grow seeds for change. The breadth and interconnectedness of the SDGs make it evident that professionals from different disciplines and sectors must work together to deliver the goals. Interdisciplinarity promotes the ability to understand complex problems like sustainability-related ones, and act on them, aligned to the expected outcomes from education for sustainable development [44,60,61]. According to the literature, interdisciplinary education has been challenging [60,62,63], and there are different ways to adopt interdisciplinarity in education for sustainable development [11,33,64]. The actions 3 to 9 in our map substantially recall this approach, but these "spot" initiatives risk being insufficient in preparing individuals to tackle complex decision-making processes in their day to day lives [11,21,65].

## 6. Conclusions

Universities play a crucial role in the implementation of sustainable development goals (SDGs) for educating the leaders of tomorrow about sustainability using new approaches. If we consider education for sustainability (EfS) a societal learning process [66], universities should be at the forefront of this effort, given that universities are supposed to be learning-centred organisations. However, universities should first learn to transcend rigid disciplinary boundaries. The field of EfS has developed, as is evident in the recent literature, and a body of literature on strategies for its implementation has emerged, disregarding the embedded, uncertain and context-related nature of SDGs' efficient and effective implementation.

Therefore, the main purpose of this paper is to organize and describe a set of ongoing education for sustainability strategies that took place from 2016 to 2019 in Italian universities. Eighteen best practices have been collected after a national call by the Italian Network of Sustainable Universities (RUS), that aimed to map the current landscape of SDGs-related actions.

We presented the analysis of 18 self-selected case studies after a "call for best practice in Sustainability Education" in 2017 across all RUS members. We filtered and described the reported activities according to declared goals and approaches by each university, so as to read the elements of governance, curricula, contents and methods that aim to integrate sustainability aspects.

Table 3 describes the characteristics of the 18 universities of the self-selected sample. Table 2 summarised the data after their qualitative analysis, where we collected, systematised and reorganised recurrent features dividing declared and inferred data according to specific categories (university name, number of students, type, the title of the initiative, EU funding, goals and level, approach, urban outreach, sustainable development goals (SDGs), driver, the mission of the initiative), drawing from the literature review and the RUS' survey structure.

We metaphorically depict the current structure of higher education in the Italian system like a city, where streets are the undergraduate/graduate programmes. Buildings of various type (big/small houses, industries, blocks, temples) are connected by roads and represent the training and educational experiences that students have to pass through. The natural elements help to identify the relationship among these experiences (for example, mountains represent a clear separation between two academic programmes, while other programmes are represented in flat terrain, symbolizing an easier possibility of exchange through a greenfield). This representation does not refer to a particular set of programmes promoted by a specific athenaeum, rather, it collects the most common characteristics of the Italian programmes. The representation is thus to be considered qualitative rather than quantitative: each object represents a type of activity and the number of objects/experiences represent their uniqueness (one object) or repetition (more than one object) within a typical academic programme. We then place, in such a map, the initiatives related to EfS as they happen in various aspects and functions at the Italian universities, according to the RUS survey.

Within the Italian higher education system's EfS efforts, a predominant mission (teaching) and a prevalent driver (top-down) have been found as the most frequent features of SDGs educational initiatives. Secondarily, sustainability is seen most often as a separate discipline to be inserted into existing curricula and original teachings, or as a conceptual tool for specific societal challenges through spot initiatives like workshops or fieldworks.

The analysis of the Italian EfS experiences gathered through the RUS national call allowed us to map certain recurrent features of initiatives taking place within the current educational structure.

In this work, we presented the first recomposition of current SD implementation strategies in Italian HEIs, highlighting in a synchronic map the methods, tools and loci of change of the Italian RUS members that responded to the call for EfS best practices. The representation of the state of the art of the Italian universities structure is symbolically a map, where the current structure and a possible transformation are no longer isolated, and are categorised according to transferability and scalability criteria.

With this paper, we do not propose a total reorienting or a solutions decalogue disregarding the local factors and the local resources in Italian universities. On the contrary, we draw a map to report how the existing structure can welcome EfS initiatives with adjustments, retrofitting actions and renewal, hopefully paving the path towards more holistic and coordinated sustainability efforts.

The map is the first country-wide systematization of the Italian higher education institutions toward SDGs implementation, to avoid individual experiences remaining isolated and self-referential, and most importantly, to provide comparability and transferability criteria to help similar cases be networked both within similar governance levels and within methodological practices. A network of universities for EfS can be a platform for knowledge sharing, presentations on strategic issues, peer to peer support, joint project proposals, debates, the creation of policy positions, the sharing of scientific intelligence and research infrastructure, networking (both personal and institutional) and, finally, a platform for following and influencing policy affairs [67].

The role of cultural and socio-economic differences [68] and of socio-economic performances of universities [66] are certainly further tracks of researches for understating what could be local obstacles for structured curricula reform actions. Of course, the actual achievement of EfS cannot depend solely on new regulatory and/or scientific stimuli, but is related to the degree of diffusion of an effective culture of responsibility within the university and the ability to legitimize the actions and epistemologies, by providing effective tools and solutions for corporate governance issues. Similarly, the management characteristics, the availability of resources, and the structures and culture of the university system, affect the governance development level and therefore the ways in which EfS is enacted [69].

Future works [34], future works may explore pedagogical and strategic tools used among European higher university institutions, as well as the enhancement of stimuli for a personal and societal transformation generated by the partnership of all those people and institutions engaged in the exciting yet urgent work to address today's societal challenges.

**Author Contributions:** All authors have read and agreed to the published version of the manuscript. The paper structure, contents framing and theoretical set were put in place by G.S. C.B. and D.C. have been in charge of the data elaboration and visualization. The all three authors were co-writing of the introduction, methodology, results and conclusions paragraphs.

**Funding:** This research was supported by TrUST—Transdisciplinarity for Urban Sustainability Transition. TrUST is a research project that aims at a better understanding of how to achieve more efficient and effective inter/trans-disciplinary research and education for urban sustainability transitions. It has been funded by the Interuniversity Department of Regional & Urban Studies and Planning, Politecnico di Torino and Università di Torino, Viale Mattioli, 39, 10,125 Turin. The APC was funded by TrUST and by authors' individual starting grants.

**Conflicts of Interest:** The authors declare no conflict of interest.

## Abbreviations

The following abbreviations are used in this manuscript:

| | |
|---|---|
| AIESEC | Association Internationale des Etudiants en Sciences Economiques et Commerciales |
| ASvis | Alliance for Sustainable Development |
| CRUI | Conferenza dei Rettori delle Università Italiane—Italian Rectors' board |
| EfS | Education for Sustainability |
| ECTS | European Credit Transfer and Accumulation System |
| HEIs | Higher Education Institutions |
| POLITO | Politecnico di Torino |
| RUS | Rete Italiana Università per la Sostenibilità—Italian Network of Sustainable Universities |
| SD | Sustainable Development |
| SDGs | Sustainable Development Goals |
| SSH | Social Science and Humanities |
| STEM | Science, Technology, Engineering and Mathematics |

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
