# Peer review of "Education for Sustainability in Practice: A Review of Current Strategies within Italian Universities"

_sustainability, doi:10.3390/su12135246_

Round 1

Reviewer 1 Report

The study focuses on the level of compliance and contribution of the SDGs in Italian higher education educational contexts. An interesting national systematization of 18 experiences is carried out to describe a first panorama of actions related to the SDGs.

However, it is recommended to incorporate a section that analyzes the results obtained from an international comparative perspective. This section will help to adequately contextualize the strengths and limitations identified in the context in which the research is focused.

Likewise, it is suggested to extend the conclusions of the study highlighting, in more detail, the educational and public implications of the results.

Author Response

Within this rebuttal letter, we would like to respond to each of the reviewers' comments and generic concerns. The resubmitted version has undergone a massive re-editing, avoiding the use of comments in the manuscript in order to address changes, but submitting this rebuttal letter in which all the suggestions have been answered one by one. We also submit two versions: one with track changes highlighted, and another without any tracked change for more comfortable reading. Our answers are in italics below.

Academic Editor Comments

Dear authors,

You have a good start here, and I would like to see this article move forward to peer review. However, this piece needs further development before moving forward. 

Thanks, Tina. Your encouragement is fundamental for us young researchers on the theme. We are keen to engage in further work.
Your abstract does not seem to entirely match your article about giving an accurate summary of it. Your paragraph at the end of your introduction seems to do a better job of outlining the article content and processes.

Thanks: we entirely rewrote it on the basis of the structure announced at the end of the intro and after completing all the review process.

Your methodology and methods need to be clarified and discussed in more depth. Tell us more about hermeneutics and grounded theory and exactly how you applied each in your study. I am quite familiar with both, and I can't tell exactly how you used them. Knowing exactly what you did and why it is important for the validity of your findings.

Our study wants to avoid the risk of the nth quantitative analysis disregarding context factors affecting the efficacy of EfS initiatives, adopting forms of grounded theory and hermeneutic phenomenology arising from the same ontological foundation, whether this is linked to a perceived view of knowledge or to a received view of knowledge.

The main purpose of this paper is, in fact, to organise and describe a set of education for sustainability strategies at the level of the organisational structure of higher education institutions.

Our research is based on an extensive and rigorous review of the most recent and most cited papers and books about sustainability and SDGs embedding into higher education curricula. In a second phase, we then developed an Excel spreadsheet (Table 2) and a graphic report (Fig. 8) as means to interpret and systematise actions, loci of change and drivers as found in the sample of the 18 Universities of our country-wide case study. We coupled empirical grounding and hermeneutics induction, to emphasise our findings from both literature review categories and the ones emerging in our sample.

Your findings need much more discussion. The vast table you provide does not explain itself. You need to tell your reader what you have found to be significant here and why. You also need to draw strong connections between the EWG material you present and your other results. These two items are placed next to each other but are hardly explained with regard to their connection.

We discussed more the analysis of 18 self-selected case studies after a "call for best practice in Sustainability Education" in 2017 across all RUS members. We filtered and described the reported activities according to declared goals and approaches so that we can read the elements of university governance, curricula, contents and methods toward further integration of sustainability aspects.

The tables in the paper summarise and systematise the information extracted from the calls of each University. We now split the data into two parts so that they are more readable.

The first part of the table (Table 2a) describes the characteristics of the 18 universities of the self-selected sample. Universities themselves declared data, but where information was missing, the authors searched on their institutional website info like the number of students, main disciplines focus, type of initiative as declared at the time of the survey, main objectives and the presence of European funds.

The second part of the table (Table 2b) summarised the data after their qualitative analysis and organised according to the induced categories from the literature review, as described in the methods paragraph.

In general, please open each section of the article with a statement about what its purpose is relative to the more extensive article. Also, close each section by providing a transition between the ideas you were just working with and the ideas you will be discussing in the following section. Doing these things will help your reader understand your logical flow of ideas.

Thanks, we did this, and it helped us to re-collocate some part of the discourse were more appropriate.

Please also use the full article template for the journal that includes line numbers. You can simply copy and paste the material you have into the template. If you need to use different headings than those provided in the template, please feel free to do so. Just change the ones present in the template to fit your work.

We did it.

I encourage you to revise and resubmit this work. It is on a very promising subject, and you use a methodological framing that could be quite useful to your purposes. I hope to see your revised resubmission soon.

Thank you very much!

Best, Tina

Open Review1

Comments and Suggestions for Authors

The study focuses on the level of compliance and contribution of the SDGs in Italian higher education educational contexts. An interesting national systematisation of 18 experiences is carried out to describe the first panorama of actions related to the SDGs. However, it is recommended to incorporate a section that analyses the results obtained from an international comparative perspective. This section will help to adequately contextualise the strengths and limitations identified in the context in which the research is focused.

Our map is atypic, looking at the lack of levels of hierarchies, meaning that a holistic effort toward a systemic sustainability shift is far away from its beginning.

The main gap we envisage with the International literature on how to embed Education for Sustainability in current Higher Education Institutions' structures and infrastructures is the lack of an inter/trans-disciplinary literacy as the preliminary ground to grow seeds for change. In fact, the breadth and interconnectedness of the SDGs make it evident that professionals from different disciplines and sectors must work together to deliver the goals. Interdisciplinarity promotes the ability to understand complex problems like sustainability-related ones, and act on them, aligned to the expected outcomes from education for sustainable development [26,54,55]. According to the literature, interdisciplinarity education has been challenging [56,57], and there are different ways to adopt interdisciplinarity in education for sustainable development (examples of practices in [29,58–62]). But although the benefits of interdisciplinarity are known, in general, it depends on the students, sometimes prompted by their teachers, to adopt a perspective that considers social, economic and environmental aspects.

The actions from 3 to 9 as numbered in our map substantially recall this approach. Embedding sustainable development only in environmental courses, or creating specific disciplines not connected to the mainstream curriculum will not be sufficient to prepare individuals to make the necessary decisions in their day to day life to address sustainability challenges [5,20,63].

Moreover, compared to the actions highlighted by the Network of the Sustainable Development Solutions Network (SDSN) Australia/Pacific (SDSN, 2018), and the relevant literature summing world-wide trends of EfS embedding in current curricula, we position Italy in the following comparative six axes: 

  • weak in the inclusion of SDGs into all undergraduate and graduate courses, as well as graduate research training [11–13];
  • absent in delivering training on SDGs to all curriculum developers, course coordinators and professors [14–16], with shy action of free-online courses dedicated to SDGs literacy;
  • on track in offering executive education and capacity building courses for external stakeholders based on SDGs [17–19], as in the actions 7, 10, 11 and 12 on our map;
  • on track in defending the implementation of national and public education policies that support education for SDGs [7,20,21], as the educational working group of the RUS is collaborating with the Italian Ministry for setting the ground infrastructure (level 0 in the map) for cultivating the ground and maybe mandatory actions for fertilising and sprout more and more sustainable seeds, and hopefully, make it rain;
  • quite good in involving students in the co-creation of learning environments that sustain learning on SDGs [22–24], as in the actions 4 and 7 in the map;
  • on track in developing courses directed to real-world collaborative projects for change [25–27], as in the actions 6 and 10 in the map.

Likewise, it is suggested to extend the conclusions of the study highlighting, in more detail, the educational and public implications of the results.

Thanks. We added reflections on the educational and public implication of our results in the conclusions. If we consider Education for Sustainability (EfS), a societal learning process [64], universities should be at the forefront of this, given that universities are supposed to be learning-centred organisations. However, universities should first unlearn to be learning organisations themselves and be able to transcend the rigid disciplinary fences. This sounds like a pure utopia, but the "Fridays for Futures" movements are telling to current education institutions that an emerging future is at the door, and that a new global university and school is in the making. That new school is characterised by "institutional inversion", where learners leave the classroom and engage with the significant hotspots of societal innovation in their cities, regions, and ecosystems. EfS in practice is a quite new, complex socio-technical phenomenon: while plural perspectives on EfS are encouraged both by practitioners and researchers, there is also a danger that such pluralism may sustain dominant political ideologies and consolidated corporate power that underprivilege ecocentric perspective, or that disregard significant differences at the 'grass-root' level of practice of EfS – both as far as goals and orientation, as well as level of educational programmes.

Reviewer 2 Report

General comment: This paper explains the implementation of SDG for Italian universities.
Introduction: The Introduction should be improve focusing more on the aim of the paper, main methods, main results and few recommendations based on empirical results. The authors should explain more their research novelty compared to previous studies from literature.
Methodology: Indicate limits and advantages of methods. Indicate alternative methods. Provide practical comments to introduce the methods.
Results: More comments of the results are required and more comparisons with similar studies from literature. More details on data are required.
Discussion: Interpretations of the results are not enough and a more critical position is required.
Bibliography/References: The reference list is not enough and not up-to-date. Add recent references, especially those from journals indexed in international databases, WoS and Scopus.
Decision: Accept with corrections.

Author Response

Within this rebuttal letter, we would like to respond to each of the reviewers' comments and generic concerns. The resubmitted version has undergone a massive re-editing, avoiding the use of comments in the manuscript in order to address changes, but submitting this rebuttal letter in which all the suggestions have been answered one by one. We also submit two versions: one with track changes highlighted, and another without any tracked change for more comfortable reading. Our answers are in italics below.

Academic Editor Comments

Dear authors,

You have a good start here, and I would like to see this article move forward to peer review. However, this piece needs further development before moving forward. 

Thanks, Tina. Your encouragement is fundamental for us young researchers on the theme. We are keen to engage in further work.
Your abstract does not seem to entirely match your article about giving an accurate summary of it. Your paragraph at the end of your introduction seems to do a better job of outlining the article content and processes.

Thanks: we entirely rewrote it on the basis of the structure announced at the end of the intro and after completing all the review process.

Your methodology and methods need to be clarified and discussed in more depth. Tell us more about hermeneutics and grounded theory and exactly how you applied each in your study. I am quite familiar with both, and I can't tell exactly how you used them. Knowing exactly what you did and why it is important for the validity of your findings.

Our study wants to avoid the risk of the nth quantitative analysis disregarding context factors affecting the efficacy of EfS initiatives, adopting forms of grounded theory and hermeneutic phenomenology arising from the same ontological foundation, whether this is linked to a perceived view of knowledge or to a received view of knowledge.

The main purpose of this paper is, in fact, to organise and describe a set of education for sustainability strategies at the level of the organisational structure of higher education institutions.

Our research is based on an extensive and rigorous review of the most recent and most cited papers and books about sustainability and SDGs embedding into higher education curricula. In a second phase, we then developed an Excel spreadsheet (Table 2) and a graphic report (Fig. 8) as means to interpret and systematise actions, loci of change and drivers as found in the sample of the 18 Universities of our country-wide case study. We coupled empirical grounding and hermeneutics induction, to emphasise our findings from both literature review categories and the ones emerging in our sample.

Your findings need much more discussion. The vast table you provide does not explain itself. You need to tell your reader what you have found to be significant here and why. You also need to draw strong connections between the EWG material you present and your other results. These two items are placed next to each other but are hardly explained with regard to their connection.

We discussed more the analysis of 18 self-selected case studies after a "call for best practice in Sustainability Education" in 2017 across all RUS members. We filtered and described the reported activities according to declared goals and approaches so that we can read the elements of university governance, curricula, contents and methods toward further integration of sustainability aspects.

The tables in the paper summarise and systematise the information extracted from the calls of each University. We now split the data into two parts so that they are more readable.

The first part of the table (Table 2a) describes the characteristics of the 18 universities of the self-selected sample. Universities themselves declared data, but where information was missing, the authors searched on their institutional website info like the number of students, main disciplines focus, type of initiative as declared at the time of the survey, main objectives and the presence of European funds.

The second part of the table (Table 2b) summarised the data after their qualitative analysis and organised according to the induced categories from the literature review, as described in the methods paragraph.

In general, please open each section of the article with a statement about what its purpose is relative to the more extensive article. Also, close each section by providing a transition between the ideas you were just working with and the ideas you will be discussing in the following section. Doing these things will help your reader understand your logical flow of ideas.

Thanks, we did this, and it helped us to re-collocate some part of the discourse were more appropriate.

Please also use the full article template for the journal that includes line numbers. You can simply copy and paste the material you have into the template. If you need to use different headings than those provided in the template, please feel free to do so. Just change the ones present in the template to fit your work.

We did it.

I encourage you to revise and resubmit this work. It is on a very promising subject, and you use a methodological framing that could be quite useful to your purposes. I hope to see your revised resubmission soon.

Thank you very much!

Best, Tina

Open Review1

Comments and Suggestions for Authors

The study focuses on the level of compliance and contribution of the SDGs in Italian higher education educational contexts. An interesting national systematisation of 18 experiences is carried out to describe the first panorama of actions related to the SDGs. However, it is recommended to incorporate a section that analyses the results obtained from an international comparative perspective. This section will help to adequately contextualise the strengths and limitations identified in the context in which the research is focused.

Our map is atypic, looking at the lack of levels of hierarchies, meaning that a holistic effort toward a systemic sustainability shift is far away from its beginning.

The main gap we envisage with the International literature on how to embed Education for Sustainability in current Higher Education Institutions' structures and infrastructures is the lack of an inter/trans-disciplinary literacy as the preliminary ground to grow seeds for change. In fact, the breadth and interconnectedness of the SDGs make it evident that professionals from different disciplines and sectors must work together to deliver the goals. Interdisciplinarity promotes the ability to understand complex problems like sustainability-related ones, and act on them, aligned to the expected outcomes from education for sustainable development [26,54,55]. According to the literature, interdisciplinarity education has been challenging [56,57], and there are different ways to adopt interdisciplinarity in education for sustainable development (examples of practices in [29,58–62]). But although the benefits of interdisciplinarity are known, in general, it depends on the students, sometimes prompted by their teachers, to adopt a perspective that considers social, economic and environmental aspects.

The actions from 3 to 9 as numbered in our map substantially recall this approach. Embedding sustainable development only in environmental courses, or creating specific disciplines not connected to the mainstream curriculum will not be sufficient to prepare individuals to make the necessary decisions in their day to day life to address sustainability challenges [5,20,63].

Moreover, compared to the actions highlighted by the Network of the Sustainable Development Solutions Network (SDSN) Australia/Pacific (SDSN, 2018), and the relevant literature summing world-wide trends of EfS embedding in current curricula, we position Italy in the following comparative six axes: 

  • weak in the inclusion of SDGs into all undergraduate and graduate courses, as well as graduate research training [11–13];
  • absent in delivering training on SDGs to all curriculum developers, course coordinators and professors [14–16], with shy action of free-online courses dedicated to SDGs literacy;
  • on track in offering executive education and capacity building courses for external stakeholders based on SDGs [17–19], as in the actions 7, 10, 11 and 12 on our map;
  • on track in defending the implementation of national and public education policies that support education for SDGs [7,20,21], as the educational working group of the RUS is collaborating with the Italian Ministry for setting the ground infrastructure (level 0 in the map) for cultivating the ground and maybe mandatory actions for fertilising and sprout more and more sustainable seeds, and hopefully, make it rain;
  • quite good in involving students in the co-creation of learning environments that sustain learning on SDGs [22–24], as in the actions 4 and 7 in the map;
  • on track in developing courses directed to real-world collaborative projects for change [25–27], as in the actions 6 and 10 in the map.

Likewise, it is suggested to extend the conclusions of the study highlighting, in more detail, the educational and public implications of the results.

Thanks. We added reflections on the educational and public implication of our results in the conclusions. If we consider Education for Sustainability (EfS), a societal learning process [64], universities should be at the forefront of this, given that universities are supposed to be learning-centred organisations. However, universities should first unlearn to be learning organisations themselves and be able to transcend the rigid disciplinary fences. This sounds like a pure utopia, but the "Fridays for Futures" movements are telling to current education institutions that an emerging future is at the door, and that a new global university and school is in the making. That new school is characterised by "institutional inversion", where learners leave the classroom and engage with the significant hotspots of societal innovation in their cities, regions, and ecosystems. EfS in practice is a quite new, complex socio-technical phenomenon: while plural perspectives on EfS are encouraged both by practitioners and researchers, there is also a danger that such pluralism may sustain dominant political ideologies and consolidated corporate power that underprivilege ecocentric perspective, or that disregard significant differences at the 'grass-root' level of practice of EfS – both as far as goals and orientation, as well as level of educational programmes.

Open Review2

Comments and Suggestions for Authors

General comment: This paper explains the implementation of SDG for Italian universities.

Introduction: The Introduction should be improve focusing more on the aim of the paper, main methods, main results and few recommendations based on empirical results. The authors should explain more their research novelty compared to previous studies from the literature.

Thanks. We rewrite it according to your suggestions.

Methodology: Indicate the limits and advantages of methods. Indicate alternative methods. Provide practical comments to introduce the methods.

Thanks. We answered this also following the editor’ same suggestions.

Results: More comments about the results are required and more comparisons with similar studies from the literature. More details on data are required.

Thanks. We wrote more about each diagram and we commented with two more pages the results.

Discussion: Interpretations of the results are not enough, and a more critical position is required.

Thanks. In the discussion we sum up the analysis of the best practice in education for sustainability as reported by the self-selected sample of 18 Italian Universities member of the RUS. For easier reading, we report the discussion also in a graphic representation. We metaphorically draw the current structure of higher education in the Italian system like a city (fig.7), where streets are the undergraduate/graduate programmes, temples are the unavoidable, necessary foundation of some disciplines and houses, courts, industries, fields and woods represents extra reaching points in the paths. We start from this map of the status quo to advance a hypothesis (fig.8) on how we can restructure the current system in order to embed education for sustainability in all places. We then symbolise the 12 main actions (fig. 9) undertaken by the universities in our sample in a hierarchic list to follow as a decalogue for paving the way to a shift in education for the University of the XXI century.

Bibliography/References: The reference list is not enough and not up-to-date. Add recent references, especially those from journals indexed in international databases, WoS and Scopus.

Thanks. We added recent references as collected from the source you indicated.

Reviewer 3 Report

The authors  detail the implementation of sustainable development goals in Italian universities from 2016 to 2019. While there is some potential to this work, the authors have to do a much better job of framing, introducing, and describing their research in order to make it suitable for publication in this journal. Here are the following points to be addressed:

1. The authors have not elaborated in the abstract or introduction on what SDG means. For readers who are not familiar with this term, I recommend that the authors define SDG in their title, abstract and introduction before moving on to using abbreviations.

2. The abstract is too long and detailed and must be shortened to improve readability. Only the important results must be highlighted in the abstract without so much informationl regarding their discussion.

3. In the introduction section, the authors should explicitly articulate the targeted goals of the initiative/program and how that is building on existing work and how it is different from what was previously done. 

4. There does not seem to be evidence/measurable outcomes about the success of the proposed strategies. The paper would benefit from qualitative information from survey data or descriptive statistics. 

5. The discussion section is largely lacking. The results are not thoroughly described and require much better articulation of the various graphs and figures, what questions they address and what are their implications.

6. The authors have not detailed the challenges faced/limitations of this approach. This is important as well as to describe or speculate on possible methods to minimize these.

7. Considering that this work was done from 2016-2019, it would be useful to include a timeline which highlights the sequence in which aspects of the project were implemented.

Author Response

Within this rebuttal letter, we would like to respond to each of the reviewers' comments and generic concerns. The resubmitted version has undergone a massive re-editing, avoiding the use of comments in the manuscript in order to address changes, but submitting this rebuttal letter in which all the suggestions have been answered one by one. We also submit two versions: one with track changes highlighted, and another without any tracked change for more comfortable reading. Our answers are in italics below.

Academic Editor Comments

Dear authors,

You have a good start here, and I would like to see this article move forward to peer review. However, this piece needs further development before moving forward. 

Thanks, Tina. Your encouragement is fundamental for us young researchers on the theme. We are keen to engage in further work.
Your abstract does not seem to entirely match your article about giving an accurate summary of it. Your paragraph at the end of your introduction seems to do a better job of outlining the article content and processes.

Thanks: we entirely rewrote it on the basis of the structure announced at the end of the intro and after completing all the review process.

Your methodology and methods need to be clarified and discussed in more depth. Tell us more about hermeneutics and grounded theory and exactly how you applied each in your study. I am quite familiar with both, and I can't tell exactly how you used them. Knowing exactly what you did and why it is important for the validity of your findings.

Our study wants to avoid the risk of the nth quantitative analysis disregarding context factors affecting the efficacy of EfS initiatives, adopting forms of grounded theory and hermeneutic phenomenology arising from the same ontological foundation, whether this is linked to a perceived view of knowledge or to a received view of knowledge.

The main purpose of this paper is, in fact, to organise and describe a set of education for sustainability strategies at the level of the organisational structure of higher education institutions.

Our research is based on an extensive and rigorous review of the most recent and most cited papers and books about sustainability and SDGs embedding into higher education curricula. In a second phase, we then developed an Excel spreadsheet (Table 2) and a graphic report (Fig. 8) as means to interpret and systematise actions, loci of change and drivers as found in the sample of the 18 Universities of our country-wide case study. We coupled empirical grounding and hermeneutics induction, to emphasise our findings from both literature review categories and the ones emerging in our sample.

Your findings need much more discussion. The vast table you provide does not explain itself. You need to tell your reader what you have found to be significant here and why. You also need to draw strong connections between the EWG material you present and your other results. These two items are placed next to each other but are hardly explained with regard to their connection.

We discussed more the analysis of 18 self-selected case studies after a "call for best practice in Sustainability Education" in 2017 across all RUS members. We filtered and described the reported activities according to declared goals and approaches so that we can read the elements of university governance, curricula, contents and methods toward further integration of sustainability aspects.

The tables in the paper summarise and systematise the information extracted from the calls of each University. We now split the data into two parts so that they are more readable.

The first part of the table (Table 2a) describes the characteristics of the 18 universities of the self-selected sample. Universities themselves declared data, but where information was missing, the authors searched on their institutional website info like the number of students, main disciplines focus, type of initiative as declared at the time of the survey, main objectives and the presence of European funds.

The second part of the table (Table 2b) summarised the data after their qualitative analysis and organised according to the induced categories from the literature review, as described in the methods paragraph.

In general, please open each section of the article with a statement about what its purpose is relative to the more extensive article. Also, close each section by providing a transition between the ideas you were just working with and the ideas you will be discussing in the following section. Doing these things will help your reader understand your logical flow of ideas.

Thanks, we did this, and it helped us to re-collocate some part of the discourse were more appropriate.

Please also use the full article template for the journal that includes line numbers. You can simply copy and paste the material you have into the template. If you need to use different headings than those provided in the template, please feel free to do so. Just change the ones present in the template to fit your work.

We did it.

I encourage you to revise and resubmit this work. It is on a very promising subject, and you use a methodological framing that could be quite useful to your purposes. I hope to see your revised resubmission soon.

Thank you very much!

Best, Tina

Open Review1

Comments and Suggestions for Authors

The study focuses on the level of compliance and contribution of the SDGs in Italian higher education educational contexts. An interesting national systematisation of 18 experiences is carried out to describe the first panorama of actions related to the SDGs. However, it is recommended to incorporate a section that analyses the results obtained from an international comparative perspective. This section will help to adequately contextualise the strengths and limitations identified in the context in which the research is focused.

Our map is atypic, looking at the lack of levels of hierarchies, meaning that a holistic effort toward a systemic sustainability shift is far away from its beginning.

The main gap we envisage with the International literature on how to embed Education for Sustainability in current Higher Education Institutions' structures and infrastructures is the lack of an inter/trans-disciplinary literacy as the preliminary ground to grow seeds for change. In fact, the breadth and interconnectedness of the SDGs make it evident that professionals from different disciplines and sectors must work together to deliver the goals. Interdisciplinarity promotes the ability to understand complex problems like sustainability-related ones, and act on them, aligned to the expected outcomes from education for sustainable development [26,54,55]. According to the literature, interdisciplinarity education has been challenging [56,57], and there are different ways to adopt interdisciplinarity in education for sustainable development (examples of practices in [29,58–62]). But although the benefits of interdisciplinarity are known, in general, it depends on the students, sometimes prompted by their teachers, to adopt a perspective that considers social, economic and environmental aspects.

The actions from 3 to 9 as numbered in our map substantially recall this approach. Embedding sustainable development only in environmental courses, or creating specific disciplines not connected to the mainstream curriculum will not be sufficient to prepare individuals to make the necessary decisions in their day to day life to address sustainability challenges [5,20,63].

Moreover, compared to the actions highlighted by the Network of the Sustainable Development Solutions Network (SDSN) Australia/Pacific (SDSN, 2018), and the relevant literature summing world-wide trends of EfS embedding in current curricula, we position Italy in the following comparative six axes: 

  • weak in the inclusion of SDGs into all undergraduate and graduate courses, as well as graduate research training [11–13];
  • absent in delivering training on SDGs to all curriculum developers, course coordinators and professors [14–16], with shy action of free-online courses dedicated to SDGs literacy;
  • on track in offering executive education and capacity building courses for external stakeholders based on SDGs [17–19], as in the actions 7, 10, 11 and 12 on our map;
  • on track in defending the implementation of national and public education policies that support education for SDGs [7,20,21], as the educational working group of the RUS is collaborating with the Italian Ministry for setting the ground infrastructure (level 0 in the map) for cultivating the ground and maybe mandatory actions for fertilising and sprout more and more sustainable seeds, and hopefully, make it rain;
  • quite good in involving students in the co-creation of learning environments that sustain learning on SDGs [22–24], as in the actions 4 and 7 in the map;
  • on track in developing courses directed to real-world collaborative projects for change [25–27], as in the actions 6 and 10 in the map.

Likewise, it is suggested to extend the conclusions of the study highlighting, in more detail, the educational and public implications of the results.

Thanks. We added reflections on the educational and public implication of our results in the conclusions. If we consider Education for Sustainability (EfS), a societal learning process [64], universities should be at the forefront of this, given that universities are supposed to be learning-centred organisations. However, universities should first unlearn to be learning organisations themselves and be able to transcend the rigid disciplinary fences. This sounds like a pure utopia, but the "Fridays for Futures" movements are telling to current education institutions that an emerging future is at the door, and that a new global university and school is in the making. That new school is characterised by "institutional inversion", where learners leave the classroom and engage with the significant hotspots of societal innovation in their cities, regions, and ecosystems. EfS in practice is a quite new, complex socio-technical phenomenon: while plural perspectives on EfS are encouraged both by practitioners and researchers, there is also a danger that such pluralism may sustain dominant political ideologies and consolidated corporate power that underprivilege ecocentric perspective, or that disregard significant differences at the 'grass-root' level of practice of EfS – both as far as goals and orientation, as well as level of educational programmes.

Open Review2

Comments and Suggestions for Authors

General comment: This paper explains the implementation of SDG for Italian universities.

Introduction: The Introduction should be improve focusing more on the aim of the paper, main methods, main results and few recommendations based on empirical results. The authors should explain more their research novelty compared to previous studies from the literature.

Thanks. We rewrite it according to your suggestions.

Methodology: Indicate the limits and advantages of methods. Indicate alternative methods. Provide practical comments to introduce the methods.

Thanks. We answered this also following the editor’ same suggestions.

Results: More comments about the results are required and more comparisons with similar studies from the literature. More details on data are required.

Thanks. We wrote more about each diagram and we commented with two more pages the results.

Discussion: Interpretations of the results are not enough, and a more critical position is required.

Thanks. In the discussion we sum up the analysis of the best practice in education for sustainability as reported by the self-selected sample of 18 Italian Universities member of the RUS. For easier reading, we report the discussion also in a graphic representation. We metaphorically draw the current structure of higher education in the Italian system like a city (fig.7), where streets are the undergraduate/graduate programmes, temples are the unavoidable, necessary foundation of some disciplines and houses, courts, industries, fields and woods represents extra reaching points in the paths. We start from this map of the status quo to advance a hypothesis (fig.8) on how we can restructure the current system in order to embed education for sustainability in all places. We then symbolise the 12 main actions (fig. 9) undertaken by the universities in our sample in a hierarchic list to follow as a decalogue for paving the way to a shift in education for the University of the XXI century.

Bibliography/References: The reference list is not enough and not up-to-date. Add recent references, especially those from journals indexed in international databases, WoS and Scopus.

Thanks. We added recent references as collected from the source you indicated.

Open Review3

The authors detail the implementation of sustainable development goals in Italian universities from 2016 to 2019. While there is some potential to this work, the authors have to do a much better job of framing, introducing, and describing their research in order to make it suitable for publication in this journal. Here are the following points to be addressed:

  1. The authors have not elaborated in the abstract or introduction on what SDG means. For readers who are not familiar with this term, I recommend that the authors define SDG in their title, abstract and introduction before moving on to using abbreviations.

Thanks. We added the Sustainable Development Goals explanation alongside the paragraphs.

  1. The abstract is too long and detailed and must be shortened to improve readability. Only the important results must be highlighted in the abstract without so much informationl regarding their discussion.

Thanks. We rewrite the abstract.

  1. In the introduction section, the authors should explicitly articulate the targeted goals of the initiative/program and how that is building on existing work and how it is different from what was previously done. 

Thanks. We specified that current literature on education for SDGs (par.2) not always considers the infrastructural and practical factors affecting the success or the failure of the practices mentioned above. That is why in this paper, we articulate how the implementation of SDGs in Italian universities took place from 2016 to 2019, to understand, apart from the syllabus proclaims, what strategies are currently undergoing to implement education for sustainability in the current educational system. Eighteen best practices have been collected after a national call by the Italian Network of Sustainable Universities (RUS), that aimed to map the current landscape of SDGs-related actions. The methodology (par.3) used in this work leverages on the hermeneutics and grounded theory approaches to analyse the results of 18 experiences of SD implementation in Italian Universities. We filter and read these reported experiences (par.4) according to the educational "container" where the SDGs implementation takes place and to the different organisational scales where it happens. Par.5 presents the results of a first mapping exercise of current SD implementation strategies in Italian HEIs, highlighting the various drivers and challenges between the Italian RUS members. Conclusions (par.6) offer the opportunity to underpin scalable and transferable features of individual projects tested successfully in a small context, eventually laying the foundation for the development of a transdisciplinary educational dimension of university programs towards SDGs embedding.

  1. There does not seem to be evidence/measurable outcomes about the success of the proposed strategies. The paper would benefit from qualitative information from survey data or descriptive statistics. 

The survey did not ask for an assessment measure of success, yet we tried to compare the reported actions against international literature pieces of evidence. Further research may advance this work via in-depth interviews with relevant stakeholders.

  1. The discussion section is largely lacking. The results are not thoroughly described and require the much better articulation of the various graphs and figures, what questions they address and what are their implications.

Thanks. We rewrote the entire session following also other reviewers' suggestions.

  1. The authors have not detailed the challenges faced/limitations of this approach. This is important as well as to describe or speculate on possible methods to minimise these.

Limitation of these methodologies lies in the fact that hermeneutical explorations have the possibility of developing valid interpretations by analysing understanding [45], but that analysis is bound to the experience of the interpreter [46]. Therefore, it may be that our result only partially reflect what has been currently done in the Italian University regarding EfS. Another limit is the paradox of the hermeneutic circle, wherein the whole has to be understood from its elements and their connections with each other, yet it presupposes that to understand the individual elements the whole has to be understood [47,48]. And that is not the case for the complex scenario of Education for Sustainability implementation. On the other hand, Grounded Theory (GT) has the advantage to close the gap between theory and empirical research [49], the concerns over the predominance of quantitative methods in social sciences, and the tendency to test existing grand theories [50]. While GT emphasises developing and building theory from data and observations [51,52], one limitation is that we can not be sure that causal connections we detect for EfS to be implemented may encompass another variable than the ones collected in our specific sample [53].

We used hermeneutics and GT to set the initial framework for the analyses of (1) actions for EfS embedding from literature reviews case studies and (2) the results of an open call for sustainability education best practices in Italian HEIs. The analyses were done on our interpretations of how different approaches and infrastructural settings are related to the level of SD implementation and how they can be scaled and moved into other similar contexts. Overall, in this paper we recognise is GT and hermeneutics in tandem are not simply used for analysing qualitative data but that they inform all aspects of the design and implementation of the study, useful when investigating topics that are under-researched and where a call and a sample are so far the only data available to start.

  1. Considering that this work was done from 2016-2019, it would be useful to include a timeline which highlights the sequence in which aspects of the project were implemented.

Unfortunately, we had to refer to data made available by RUS after their survey, and not every case reported the year as requested in the questionnaire.

Round 2

Reviewer 1 Report

Suggestions and recommendations have been addressed.

Author Response

Thanks!

Reviewer 2 Report

Accept in present form

Author Response

thanks!

Reviewer 3 Report

The authors have made substantial revisions to the manuscript and I find it acceptable in its current form. Here are some minor edits:

  1. In title 'a' in 'a review' must be capitalized
  2. Introduction Line 39: What does '2030 Agenda' refer to? Please elaborate and provide proper reference for that statement
  3. Results and Discussion Line 270: Authors present analysis of "18 self-selected case studies". What were the criteria for this self-selection? Please include details regarding that.
  4. Note for editor: The article requires further editing from the journal team in order to rectify English grammar and sentence structure anomalies.

Author Response

In title 'a' in 'a review' must be capitalized

Done!

Introduction Line 39: What does '2030 Agenda' refer to? Please elaborate and provide proper reference for that statement

Thanks. We added the following reference and explanation: “The Sustainable Development Goals are a universal call to action to end poverty, protect the planet and improve the lives and prospects of everyone, everywhere. The 17 Goals were adopted by all UN Member States in 2015, as part of the 2030 Agenda for Sustainable Development which set out a 15-year plan to achieve the Goals [1]. While the UN 2030 Agenda is a plan of action for people, planet and prosperity, it also seeks to strengthen universal peace in larger freedom. represents an excellent opportunity for the change demanded by the entire society, the target 4.7 is specifically related to Education for Sustainable Development (ESD)”

Results and Discussion Line 270: Authors present analysis of "18 self-selected case studies". What were the criteria for this self-selection? Please include details regarding that.

Thanks. We developed the explanation of the selected cases with the following: “We now present the results of the analysis of the 18 case studies come out of a national "call for best practice in Sustainability Education" in 2017 across all RUS members. The 18 cases result from a collection of reporting cards among the Education Working Group (EWG) of the Italian Network of Sustainable Campuses (RUS), in its first year of activity, i.e. 2018. EWG focuses on the various approaches to education for sustainable development, highlighting good practices and proposing new ways to ensure that all university students know about the 2030 Agenda and the principles of sustainable development. We filtered and described the reported cards according to declared goals and approaches, so that the elements of university governance, curricula, contents and methods toward further integration of sustainability aspects can be read horizontally across the cases.

The tables below summarize and systematize the information extracted from the reported actions by each university.”

Note for editor: The article requires further editing from the journal team in order to rectify English grammar and sentence structure anomalies.